# The complete chloroplast genome sequences of eight *Orostachys* species: Comparative analysis and assessment of phylogenetic relationships

Ha-Rim Lee[1]☉, Kyung-Ah Kim[2]☉, Bo-Yun Kim[3], Yoo-Jung Park[1], Yoo-Bin Lee[1], Kyeong-Sik Cheon🄳[1]*

1 Department of Biological Science, Sangji University, Wonju, South Korea, 2 Environmental Research Institute, Kangwon National University, Chuncheon, South Korea, 3 Plant Resources Division, National Institute of Biological Resources, Incheon, South Korea

☉ These authors contributed equally to this work.
* cheonks@sangji.ac.kr

**Data Availability Statement:** All relevant data are within the article and its Supporting Information files.

## Abstract

We analyzed the complete chloroplast genomes of eight *Orostachys* species and compared the sequences to those of published chloroplast genomes of the congeneric and closely related genera, *Meterostachys* and *Hylotelephium*. The total chloroplast genome length of thirteen species, including the eight species analyzed in this study and the five species analyzed in previous studies, ranged from 149,860 (*M. sikokianus*) to 151,707 bp (*H. verticillatum*). The overall GC contents of the genomes were almost identical (37.6 to 37.8%). The thirteen chloroplast genomes each contained 113 unique genes comprising 79 protein-coding genes, 30 tRNA genes, and four rRNA genes. Among the annotated genes, sixteen genes contained one or two introns. Although the genome structures of all *Orostachys* and *Hylotelephium* species were identical, *Meterostachys* differed in structure due to a relatively large gene block (*trnS-GCU-trnS-GGA*) inversion. The nucleotide diversity among the subsect. *Orostachys* chloroplast genomes was extremely low in all regions, and among the subsect. *Appendiculatae*, genus *Orostachys*, and all thirteen chloroplast genomes showed high values of Pi (>0.03) in one, five, or three regions. The phylogenetic analysis showed that *Orostachys* formed polyphyly, and subsect. *Orostachys* and *Appendiculatae* were clustered with *Hylotelephium* and *Meterostachys*, respectively, supporting the conclusion that each subsection should be considered as an independent genus. Furthermore, the data supported the taxonomic position of *O. margaritifolia* and *O. iwarenge* f. *magnus*, which were treated as synonyms for *O. iwarenge* in a previous study, as independent taxa. Our results suggested that *O. ramosa* and *O. japonica* f. *polycephala* were individual variations of *O. malacophylla* and *O. japonica*, respectively. The exact taxonomic position of *O. latielliptica* and the phylogenetic relationship among the three species, *O. chongsunensis*, *O. malacophylla* and *O. ramosa*, should be a topic of future study.

**Funding:** This work was supported by the National Research Foundation of Korea (NRF) grant funded by the Korea government (MSIT) [No. 2019R1G1A1004202], and the National Institute of Biological Resources (NIBR) [No. NIBR202104101].

**Competing interests:** The authors have declared that no competing interests exist.

## Introduction

The family Crassulaceae DC., belonging to Rosales Bercht. & J.Presl, includes succulent herbaceous plants. Approximately 1500 species in 35 genera are known and they are mainly distributed throughout the Northern Hemisphere [1, 2].

Among these, the genus *Orostachys* Fisch. includes approximately 20 to 25 taxa that are distributed from the Ural Mountains to Japan [3–5]. This genus has traditionally been used as an ornamental and a medicinal plant, and some species have recently been shown to be effective in antioxidant and anticancer treatments, thus becoming recognized as a very important plant resource [6, 7].

The taxa belonging this genus are mostly biennial herbaceous plants that are usually succulent. The morphological characteristics of this genus are as follows: the roots are fibrous and there is no rhizome. The leaves are linear to ovate, often with dull purple dots; the apex is usually cuspidate with a white and cartilaginous appendage that is softly obtuse or acuminate. In the first year, the leaves stand together in solitary, basal, and dense rosettes. The flowering stem arises from the center of the rosette in the second year. The inflorescence is a dense raceme or thyrse, narrowly pyramidal to cylindrical, and contains many flowers and foliage-like bracts. The flowers are bisexual, subsessile or pedicellate, and pentamerous. The sepals are usually shorter than the petals. The petals are subconnate at the base and are white, yellow-green, or pinkish to reddish [8, 9].

Due to differences in the growth habits among closely related taxa, the genus *Orostachys* was first described as a genus independent of *Coyledon* L. by Fischer [10]. However, Steudel [11] and de Candolle [12] proposed the classification of *Orostachys* within the genera *Sedum* and *Umbilicus* DC., respectively. Since then, various studies [1, 13, 14] have been conducted to support the proposal to treat *Orostachys* as an independent genus; as a result, it is currently recognized as a genus.

The genus comprised two sections, *Orostachys* Ohba and *Schoenlandia* Ohba, that differ with respect to the shape of their leaves, number of stamens, and type of inflorescence [15]. Later sect. *Schoenlandia* was classified in a new genus, *Kungia* K.T.Fu [16]. Currently, therefore, the genus *Orostachys* has only one section, *Orostachys*, and it is split into two subsections, *Appendiculatae* (Boriss.) Ohba and *Orostachys* (Boriss.) Ohba, depending on the absence or presence of an appendage at the leaf apex [9, 17].

Recently molecular phylogenetic studies [18–21] revealed a large phylogenetic distance between the two subsections of *Orostachys*; subsect. *Orostachys* was in the *Hylotelephium* Ohba clade, and subsect. *Appendiculatae* formed a clade with *Meterostachys* Nakai. Therefore, more research is necessary to clarify the proper classification of *Orostachys*. The external morphology characteristics among species of *Orostachys* are very similar. Within each taxon, there is wide variation in external morphological characteristics. For these reasons, the classification of these plants is known to be very difficult. Although several taxonomic studies [5, 6, 18–22] have been conducted, the phylogenetic relationships among the species and the taxonomic position of many taxa remain unclear.

We obtained the whole chloroplast genome sequences of eight *Orostachys* species (*O. chongsunensis* Y.N.Lee, *O. latielliptica* Y.N.Lee, *O. malacophylla* (Pall.) Fisch, *O. iwarenge* (Makino) H.Hara, *O. iwarenge* f. *magnus* Y.N.Lee, *O. japonica* f. *polycephala* (Makino) H. Ohba, *O. margaritifolia* Y.N.Lee, and *O. ramosa* Y.N.Lee), and compared the sequence to those of five published congeneric and closely related genera (*Meterostachys* and *Hylotelephium*) chloroplast genomes, i.e., those from *O. japonica* (Maxim.) A.Berger, *O. minuta* (Kom.) A.Berger, *M. sikokianus* (Makino) Nakai, *H. erythrostictum* (Miq.) H.Ohba, and *H. verticillatum* (L.) H.Ohba. The main goal of this study was to evaluate the phylogenomic

relationships among the two subsections of *Orostachys* and its closely related genera. To address the fact that taxonomists differ in their opinions regarding the classification some species, it was our goal to clarify the taxonomic position of several taxa.

## Materials and methods

### Taxon sampling, DNA extraction, sequencing, assembly and annotation

The eight *Orostachys* taxa examined in this study were not classified as endangered or protected. We did not collect materials from any privately owned or protected areas requiring permission for collection. The plant materials for this study were collected from the native habitats of each taxon, and the voucher specimens were deposited in the Sangji University Herbarium (SJUH) (S1 Table). Total DNA was extracted from approximately 100 mg of fresh leaves using a DNeasy Plant Mini Kit (Qiagen Inc., Valencia, CA, USA), and sequenced using the Illumina MiSeq and NovaSeq 6000 platforms (Illumina Inc., San Diego, CA, USA) at Labgenomics (Seongnam, Korea). The DNA of the *Orostachys* taxa was sequenced to produce 385,646–21,445,885 raw reads with lengths of 301 bp and 150 bp (S1 Table). Low-quality sequences (Phred score < 20) were trimmed using CLC Genomics Workbench (version 6.04; CLC Inc., Arhus, Denmark). Then, reads were assembled using a Geneious assembler with a medium sensitivity option via Geneious Prime v.2022.1.1 (Biomatters Ltd., Auckland, New Zealand). The draft genome contigs were merged into a single contig by joining the overlapping terminal sequences of each contig. The protein-coding genes, transfer RNAs (tRNAs), and ribosomal RNAs (rRNAs) in the chloroplast genome were predicted and annotated using Geneious Prime v.2022.1.1 and manually edited by comparison with the published chloroplast genome sequences of *Orostachys*. The tRNAs were confirmed using tRNAscan-SE [23]. A circular chloroplast genome map was drawn using the OGDRAW program [24].

### Comparative genome analyses in *Orostachys* and allied genera

The newly complete chloroplast genome sequences of eight *Orostachys* taxa were used along with the following chloroplast genome sequences from GenBank of NCBI for comparative analysis: two published *Orostachys*, *O. japonica* (accession no. MN794320) and *O. minuta* (accession no. OK094425) sequences, one *Meterostachys*, *M. sikokianus* (Makino) Nakai (accession no. MZ365442), and two *Hylotelephium*, *H. erythrostictum* (accession no. MZ519882) and *H. verticillatum* (accession no. MT558730) sequences.

The program mVISTA was used to compare similarities among the thirteen species using the shuffle-LAGAN mode [25]. The annotated *O. malacophylla* chloroplast genome was used as a reference. Additionally, the genome structure of the thirteen species were compared using the MAUVE program [26]. The large single copy/inverted repeat (LSC/IR) and inverted repeat/small single copy (IR/SSC) boundaries of these species were also compared and analyzed.

### Nucleotide diversity analysis

To assess the nucleotide diversity (Pi) among the thirteen chloroplast genomes, including ten *Orostachys*, one *Meterostachys* and two *Hylotelephium*, the complete chloroplast genome sequences were aligned using the MAFFT [27] aligner tool and manually adjusted with BioEdit [28]. We then performed sliding window analysis to calculate the nucleotide variability (Pi) values using DnaSP 6 [29] with a window length of 600 bp and a step size of 200 bp [30].

## Phylogenetic analysis

Two data sets (whole chloroplast genome sequences and 79 protein-coding gene (PCG) sequences) from 34 Crassulaceae species were compiled into a single file of size 164,887 bp and 69,392 bp, respectively, and aligned using MAFFT [27]. Thirty-two Telephium clade [21] species were selected as the ingroups, and two species from subfam. Kalanchoideae (*Cotyledon tomentosa* Harv., *Kalanchoe delagoensis* Eckl. & Zeyh.) were chosen as the outgroups (S2 Table). Maximum likelihood (ML) analyses were performed using raxmlGUI v.2.0.6 with 1000 bootstrap replicates and the GTR+I+Γ model [31]. Bayesian inference (ngen = 1,000,000, samplefreq = 200, burninfrac = 0.25) was carried out using MrBayes v3.0b3 [32], and the best substitution model (GTR+I+Γ) was determined by the Akaike information criterion (AIC) in jModeltest version 2.1.10 [33].

## Results

### Chloroplast genome features of *Orostachys* and related genera

The chloroplast genomes of eight new *Orostachys* species have been submitted to GenBank of the National Center for Biotechnology Information (NCBI) (Table 1). The total length of the chloroplast genomes of the thirteen species, i.e., the eight species analyzed in this study and the species analyzed in previous studies (*O. japonica* and *O. minuta*, *M. sikokianus*, *H. erythrostictum*, and *H. verticillatum*), ranged from 149,860 (*M. sikokianus*) to 151,707 bp (*H. verticillatum*), and among the *Orostachys* species, *O. minuta* was the smallest (150,369 bp) and *O. latielliptica* was the largest (151,462 bp) (Table 1 and Fig 1). All thirteen cp genomes exhibited the typical quadripartite structure, consisting of a pair of IR regions (25,285–25,854 bp), separated by an LSC region (82,293–83,070 bp), and an SSC region (16,839–17,018 bp). Their overall GC contents were almost identical (37.6–37.8%). The chloroplast genomes of the thirteen species contained 113 unique genes comprising 79 protein-coding genes, 30 tRNA genes, and four rRNA genes (Table 1). Among the annotated genes, fourteen genes (*atpF*, *ndhA*, *ndhB*, *petB*, *petD*, *rpl2*, *rps12*, *rpl16*, *rpoC1*, *trnA-UGC*, *trnI-GAU*, *trnK-UUU*, *trnL-UAA*, and *trnV-UAC*) contained one intron, and two genes (*clpP* and *ycf3*) contained two introns.

The pairwise cp genomic alignment among the thirteen species (ten *Orostachys*, one *Meterostachys* and two *Hylotelephium* species) were all similar, with the exception of that of *Meterostachys*.

**Table 1.** Comparison of chloroplast genome features of *Orostachys* and related genera.

| Taxa | Length (bp) | | | | %GC | No. of genes | | | | Accession No. |
|---|---|---|---|---|---|---|---|---|---|---|
| | Total | LSC | SSC | IR | | Total | PCG | tRNA | rRNA | |
| *O. chongsunensis* | 151,399 | 82,898 | 16,875 | 25,813 | 37.8 | 113 | 79 | 30 | 4 | ON979333 |
| *O. iwarenge* | 151,431 | 82,924 | 16,881 | 25,813 | 37.8 | 113 | 79 | 30 | 4 | ON979332 |
| *O. iwarenge* f. *magnus* | 151,276 | 82,784 | 16,868 | 25,812 | 37.8 | 113 | 79 | 30 | 4 | MW851201 |
| *O. japonica* | 150,464 | 83,035 | 16,859 | 25,285 | 37.7 | 113 | 79 | 30 | 4 | MW579549 |
| *O. japonica* f. *polycephala* | 150,464 | 83,035 | 16,859 | 25,285 | 37.7 | 113 | 79 | 30 | 4 | ON979327 |
| *O. latielliptica* | 151,462 | 83,004 | 16,866 | 25,796 | 37.7 | 113 | 79 | 30 | 4 | ON979328 |
| *O. malacophylla* | 151,374 | 82,872 | 16,876 | 25,813 | 37.8 | 113 | 79 | 30 | 4 | ON979331 |
| *O. margaritifolia* | 151,112 | 82,562 | 16,842 | 25,854 | 37.8 | 113 | 79 | 30 | 4 | ON979329 |
| *O. minuta* | 150,369 | 82,795 | 16,854 | 25,360 | 37.7 | 113 | 79 | 30 | 4 | OK094425 |
| *O. ramosa* | 151,424 | 82,924 | 16,874 | 25,813 | 37.8 | 113 | 79 | 30 | 4 | ON979330 |
| *M. sikokianus* | 149,860 | 82,293 | 16,879 | 25,344 | 37.6 | 113 | 79 | 30 | 4 | MZ365442 |
| *H. erythrostictum* | 151,707 | 83,070 | 17,018 | 25,793 | 37.7 | 113 | 79 | 30 | 4 | MZ519882 |
| *H. verticillatum* | 151,398 | 82,951 | 16,839 | 25,804 | 37.8 | 113 | 79 | 30 | 4 | MT558730 |

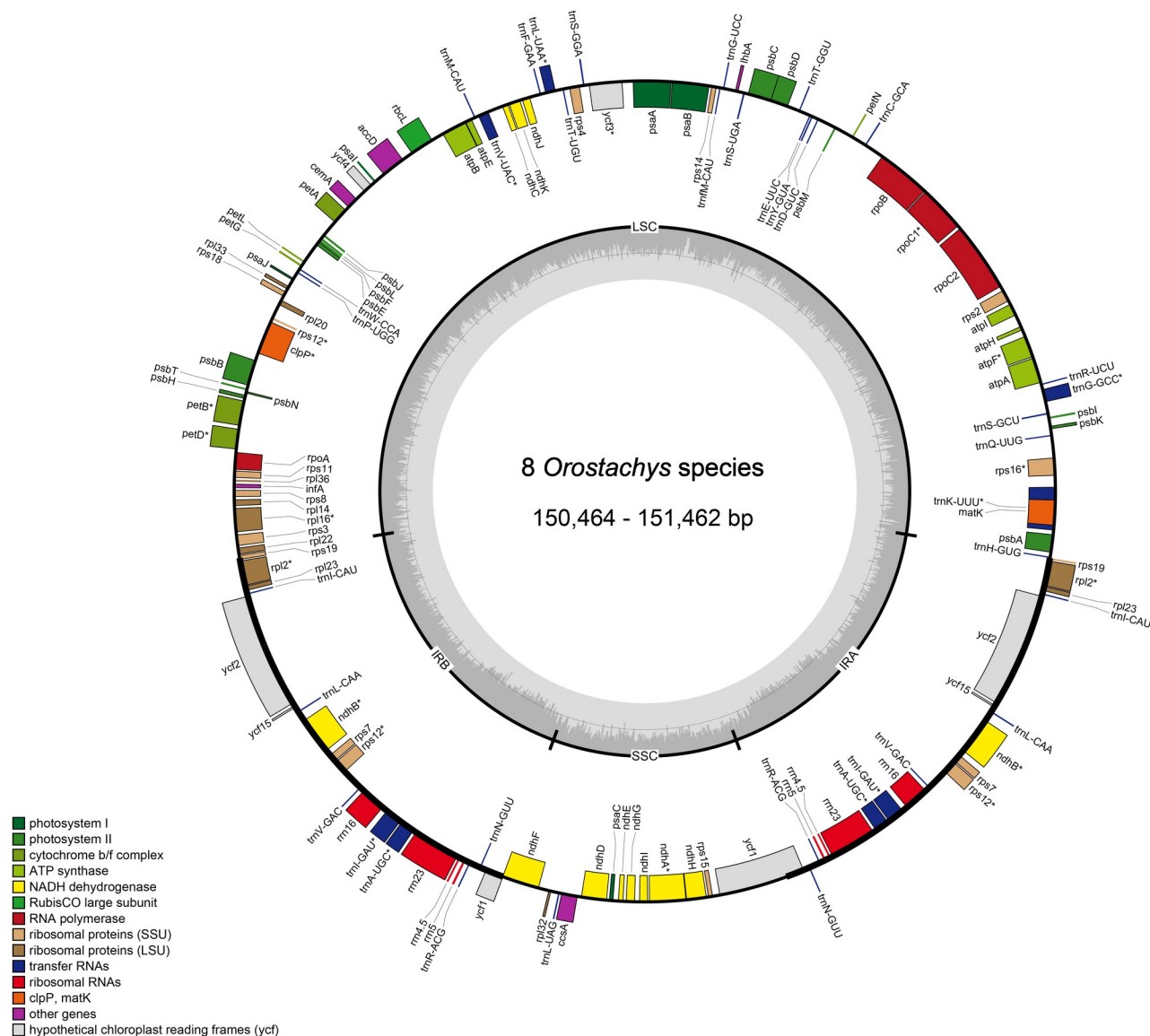

**Fig 1. Map of the newly analyzed chloroplast genome of eight *Orostachys*.** Genes inside the circle are transcribed clockwise, gene outside are transcribed counterclockwise. The dark grey inner circle corresponds to the GC content, the light gray to the AT content.

The LSC and SSC regions were more variable than the IR regions (Fig 2). In the chloroplast genome of *Meterostachys*, the sequence similarity was very low in the relatively large gene block (*trnS-GCU—trnS-GGA*, approximately 37,000 bp) of the LSC region, which was confirmed to be caused by an inversion (S1 Fig). A comparison of the LSC/IR and IR/SSC boundaries in the thirteen species is shown in Fig 3. The *rps19* gene crossed the boundary between the LSC (169 bp) and IRb (110 bp), and the *ndhF* gene and *ycf1* gene were situated in the boundary IRb (15–44 bp) and SSC (2172–2199 bp), and the boundary SSC (4038–4071 bp) and IRa (1089–1092 bp).

## Nucleotide diversity among the *Orostachys* and related genera

The average nucleotide diversity (Pi) among the five subsect. *Orostachys* species (i.e., *O. chongsunensis*, *O. malacophylla*, *O. iwarenge*, *O. iwarenge* f. *magnus*, and *O. ramosa*), five subsect.

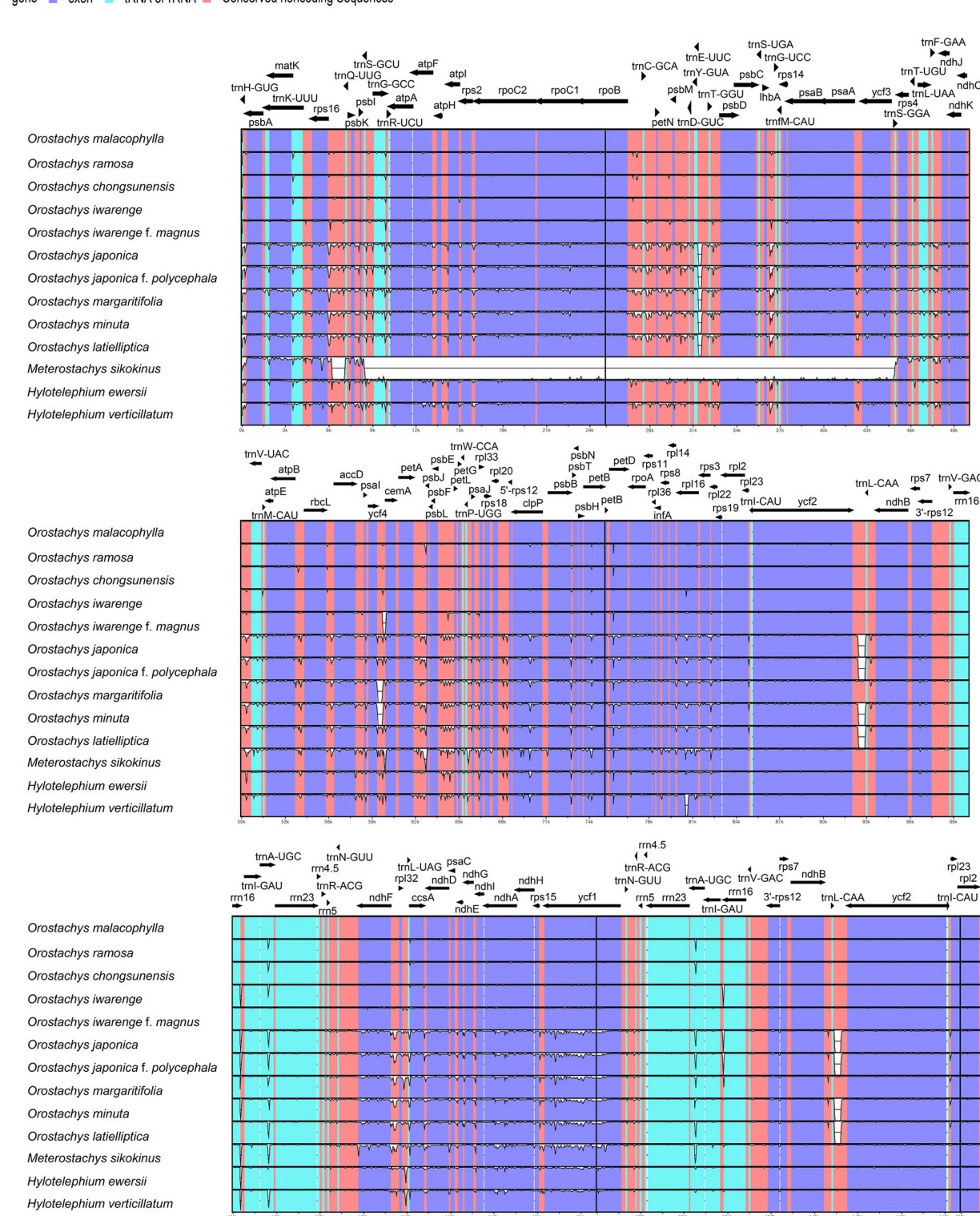

**Fig 2. Visualization of alignment of thirteen chloroplast genomes using *O. malacophylla* as a reference.** The vertical scale indicates the percent identity, ranging from 50% to 100%. Coding genes, RNAs, and non-coding regions are marked in purple, sky blue, and red, respectively.

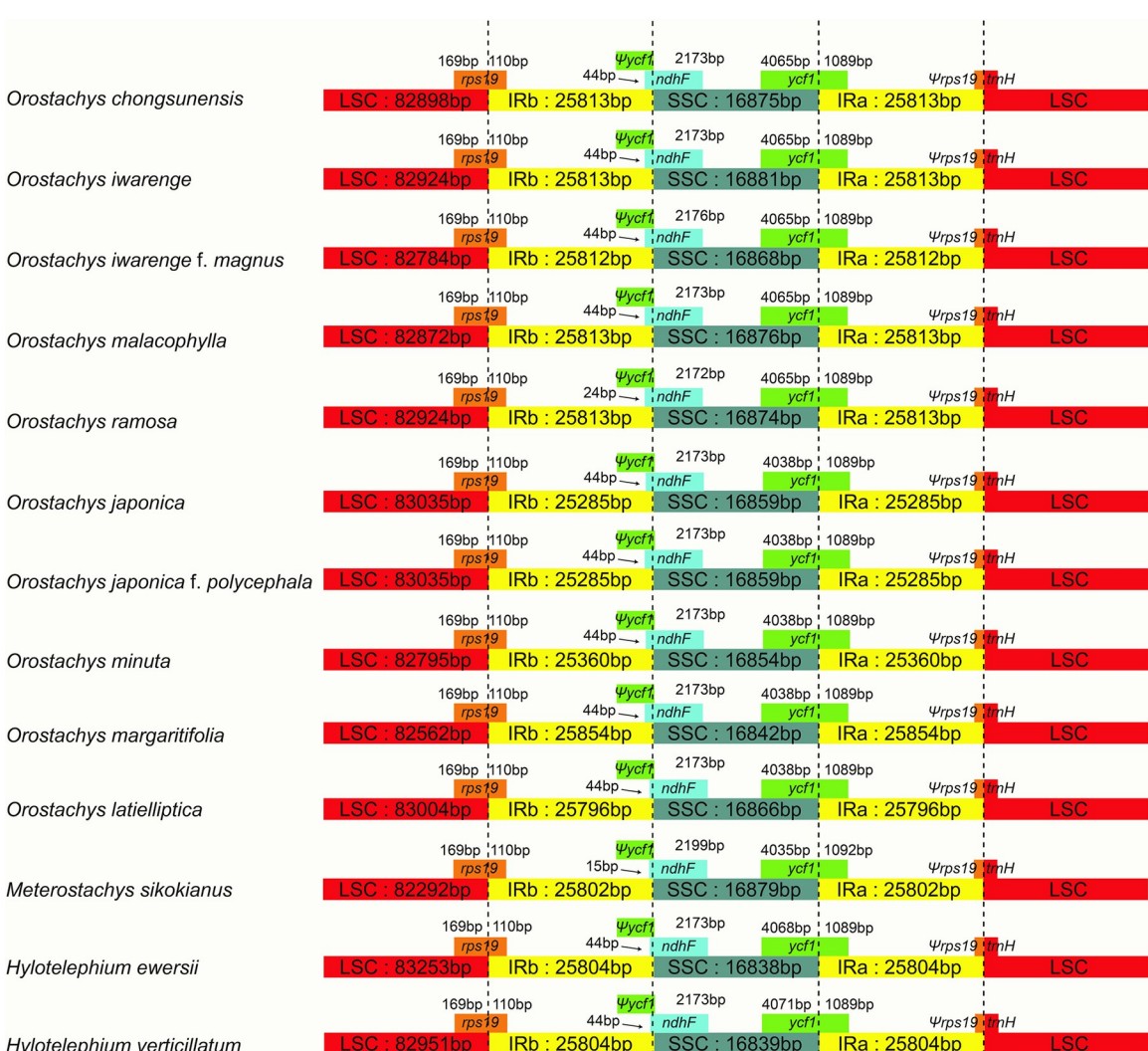

**Fig 3. Comparison of the LSC, IR, and SSC junction positions in the thirteen chloroplast genomes.**

*Appendiculatae* species (i.e., *O. japonica*, *O. japonica* f. *polycephala*, *O. latiellliptica*, *O. minuta*, and *O. margaritifolia*), ten genus *Orostachys* species, and all the cp genomes selected in this study were estimated to be 0.001, 0.002, 0.009, and 0.026, respectively. Among the five cp genomes of subsect. *Orostachys* species, the Pi values were extremely low in all regions, and the region with the highest value (*ycf4-cemA*) had a Pi value of only 0.014. Among the five cp genomes of subsect. *Appendiculatae* species, only one region (*ycf4-cemA*) showed high values of Pi (>0.03). In the ten genus *Orostachys* species, five regions (*rps16-trnQ*, *trnC-petN*, *ycf4-cemA*, *cemA*, and *ycf1*) had a high value of Pi (>0.03). In all thirteen species, three regions (*trnH-psbA*, *ycf4-cemA*, and *ycf1*) had a high value of Pi (>0.03) (Fig 4). Meanwhile, the Pi values of *rbcL* and *matK*, which are corebarcode regions, were very low, 0.005 and 0.021, respectively.

## Phylogenetic analyses of the *Telephium* clade of Crassulaceae

The two ML trees constructed based on the two data sets, whole cp genome sequences and 79 protein-coding genes, were well supported at the genus level, except for those of *Orostachys*.

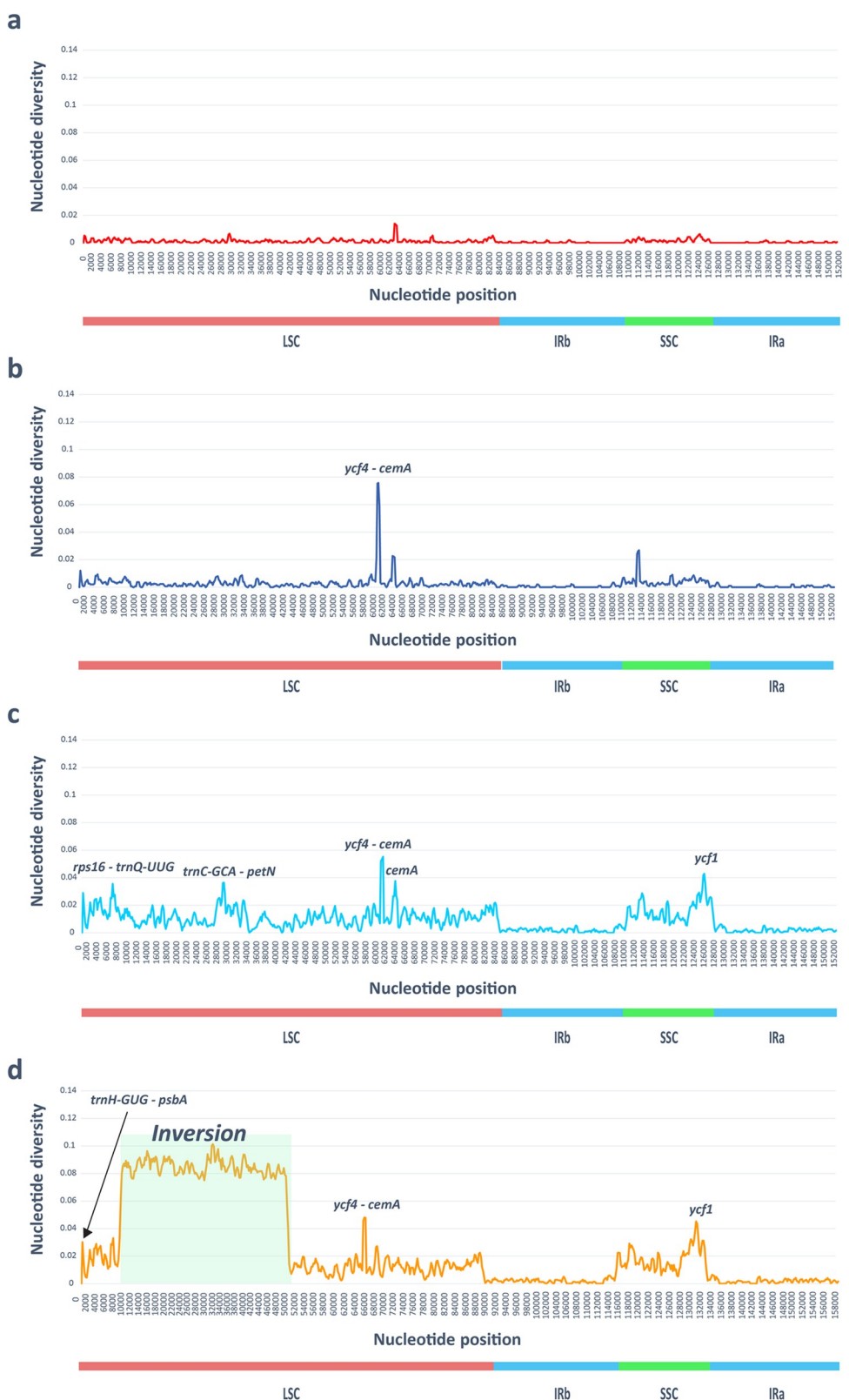

**Fig 4. Sliding window analysis of thirteen chloroplast genomes.** a; Pi values of five subsect. *Orostachys* species, b; Pi values of five subsect. *Appendiculatae* species, c; Pi values of ten genus *Orostachys* species, d; Pi values of thirteen species, including genus *Orostachys* and related genera.

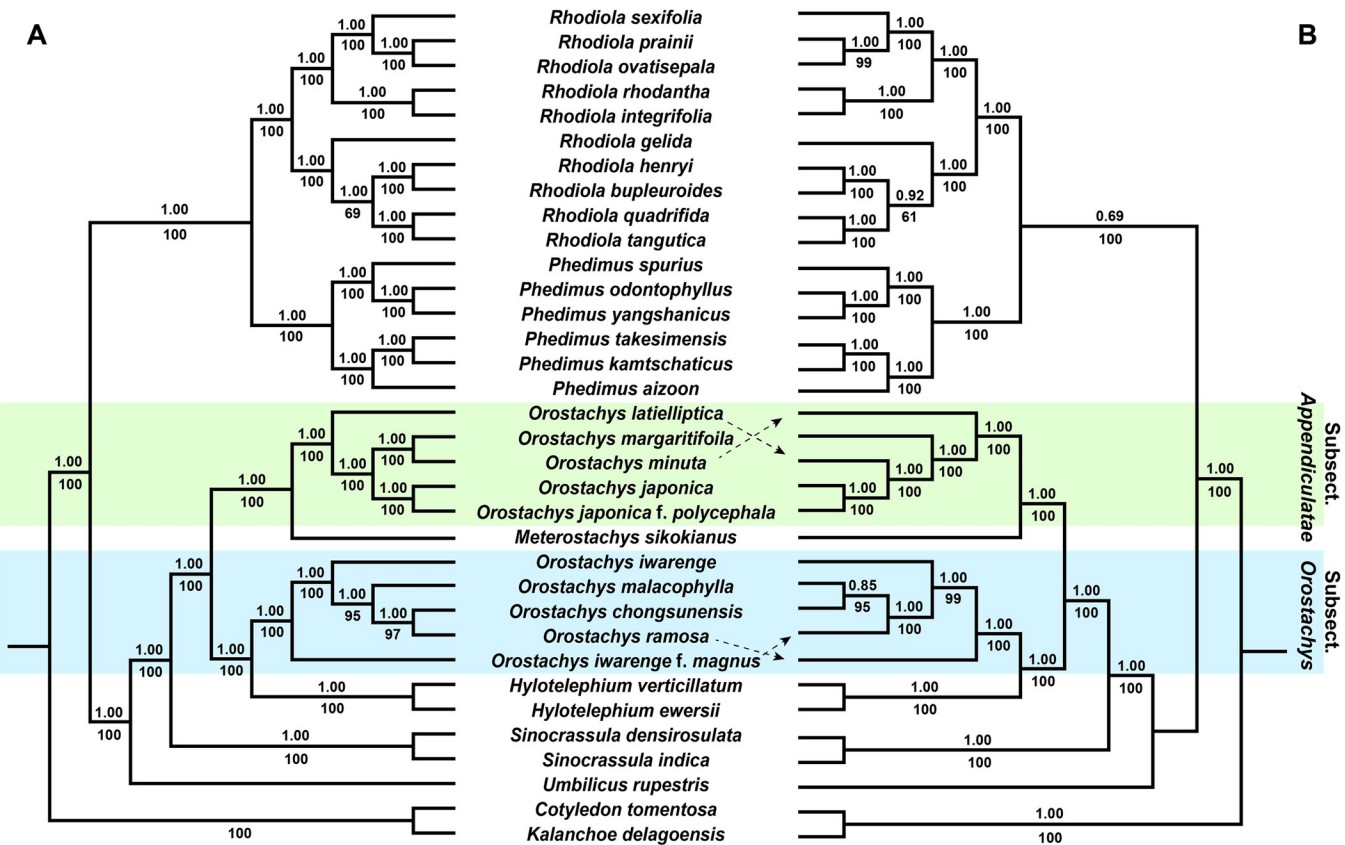

**Fig 5.** The ML trees based on whole chloroplast genome sequences (A) and 79 protein-coding genes (B) from 32 Telephium clade species and two outgroups. The bootstrap (BP) values greater than 50% are below the clades, and the Bayesian posterior probabilities (PP) are shown above the clades.

The two ML trees were divided into two subclades. The first clade consisted of *Rhodiola* L. and *Phedimus* Raf., and both genera were well supported monophyly. The second clade comprised *Orostachys*, *Meterostachys*, *Hylotelephium*, *Sinocrassula* A.Berger and *Umbilicus* DC., and *Umbilicus* formed the most basal part. *Orostachys* formed polyphyly, and subsect. *Orostachys* and *Appendiculatae* were clustered with *Hylotelephium* and *Meterostachys*, respectively (Fig 5).

In the subsect. *Orostachys* clade, *O. iwarenge* f. *magnus* formed the most basal part, followed by *O. iwarenge*, and *O. malacophylla* formed the sister to *O. ramosa* and *O. chongsunensis* in the ML tree based on whole cp genome sequences, whereas *O. ramosa* formed the sister to all other species, and *O. iwarenge* f. *magnus* formed the sister to *O. malacophylla* and *O. chongsunensis* in the ML tree based on 79 PCG sequences.

In the subsect. *Appendiculatae* clade, *O. latielliptica* was clustered at the most basal part of the tree based on whole cp genome sequences, but *O. minuta*, which formed a clade with *O. margaritifolia*, was related to all other species in the tree based on 79 PCG sequences. Additionally, the close relationship between two species, *O. japonica* and *O. japonica* f. *polycephala*, was found in both trees.

## Discussion

### Comparison of the chloroplast genomes of *Orostachys* and related genera

Many recent studies have been carried out to clarify the taxonomy of related taxa using complete cp genome sequences. The cp genome is known to be highly conserved in most land

plants, but structural changes in chloroplast genomes, such as gene duplication and deletion and inversion due to occasional rearrangements, provide important taxonomic data [34–42]. This study showed that the genome structures of *Orostachys* and *Hylotelephium* species were identical, and the sequence identities were also very similar among species in most of the chloroplast regions (Fig 2 and S1 Fig). The LSC/IRs/SSC boundaries were also very similar except for a very slight difference in sequence length (Fig 3). Therefore, these results indicated that the chloroplast genomes of *Orostachys* and *Hylotelephium* were very conservative, and that the two genera were closely related. However, the cp genome of *Meterostachys* was different in structure from those of the other taxa in this study due to the inversion of a relatively large gene block (*trnS-GCU—trnS-GGA*); this characteristic supports the classification of this group as unique genus.

## Phylogenetic relationships of *Orostachys* and related genera

In many phylogenetic studies [18–21], the genus *Orostachys*, first described by Fischer [10], was confirmed to be not monophyletic, and the two subsections, subsect. *Orostachys* and *Appendiculatae*, showed close relationships with *Hylotelephium* and *Meterostachys*, respectively. These results cast doubt on the classification of *Orostachys* to a single genus. In our study, we found that the genus was not monophyletic; we confirmed that each subsection was monophyletic and closely related to the two genera mentioned above (Fig 5). The two subsections of *Orostachys* were clearly distinguished morphologically by the absence (subsect. *Orostachys*) (Fig 6A) or presence (subsect. *Appediculatae*) (Fig 6B) of appendage such as a thorn at the leaf apex [9, 17]. Two genera, which were closely clustered with each subsection, were identified that shared this characteristic.

*Meterostachys*, which formed a clade with subsect. *Appendiculatae*, is a monotypic genus containing only one species, *M. sikokianus*. It was initially described as *Cotyledon sikokianus* Makino [43]. Later, it was segregated into a new genus [44], but a relatively recent phylogenetic study [45] suggested that it should be classified in the genus *Orostachys* because it was situated within a subsect. *Appendiculatae* clade and formed a clade with *Orostachys thyrsiflora* Fisch. *Meterostachys* is clearly distinguished from *Orostachys* by inflorescence morphology (thyrsoid-paniculate to paniculate in *Orostachys* and bracteates cymose in *Meterostachys*) (Fig 6C and 6D) [15]. We found that *Meterostachys* had a unique chloroplast genome structure and formed an independent clade in this study (Figs 2 and 5). Therefore, we think that the taxonomic position of *Meterostachys* as an independent genus is supported. Based on these results, we strongly agree with Gontcharova et al. [19] that subsect. *Appediculatae* should be recognized as a distinct genus.

Meanwhile, the phylogenetic relationships between subsect. *Orostachys* and *Hylotelephium* were not clear in previous studies [18–21]. The phylogenetic relationships examined in this study were very clear (Fig 5), but only two *Hylotelephium* species were assessed. Further studies including more diverse taxa are needed.

## Phylogenetic relationships and taxonomic position of *Orostachys* species

The phylogenetic relationships and taxonomic position of many taxa in *Orostachys* remain contested. Species, such as *O. iwarenge* f. *magnus*, *O. ramosa*, *O. latielliptica*, *O. chongsunensis*, and *O. margaritifolia*, first described by Lee and Lee [8, 46], are particularly problematic in terms of their taxonomic position because Ohba [17] treated these taxa as synonyms for *O. iwarenge*, *O. japonica*, and *O. malacophylla*, without taxonomic studies.

In all ML trees obtained in this study (Fig 5), *O. margaritifolia* (Fig 6E), which was treated as synonym for *O. iwarenge* (Fig 6F) by Ohba [17], was not clustered within the subsect.

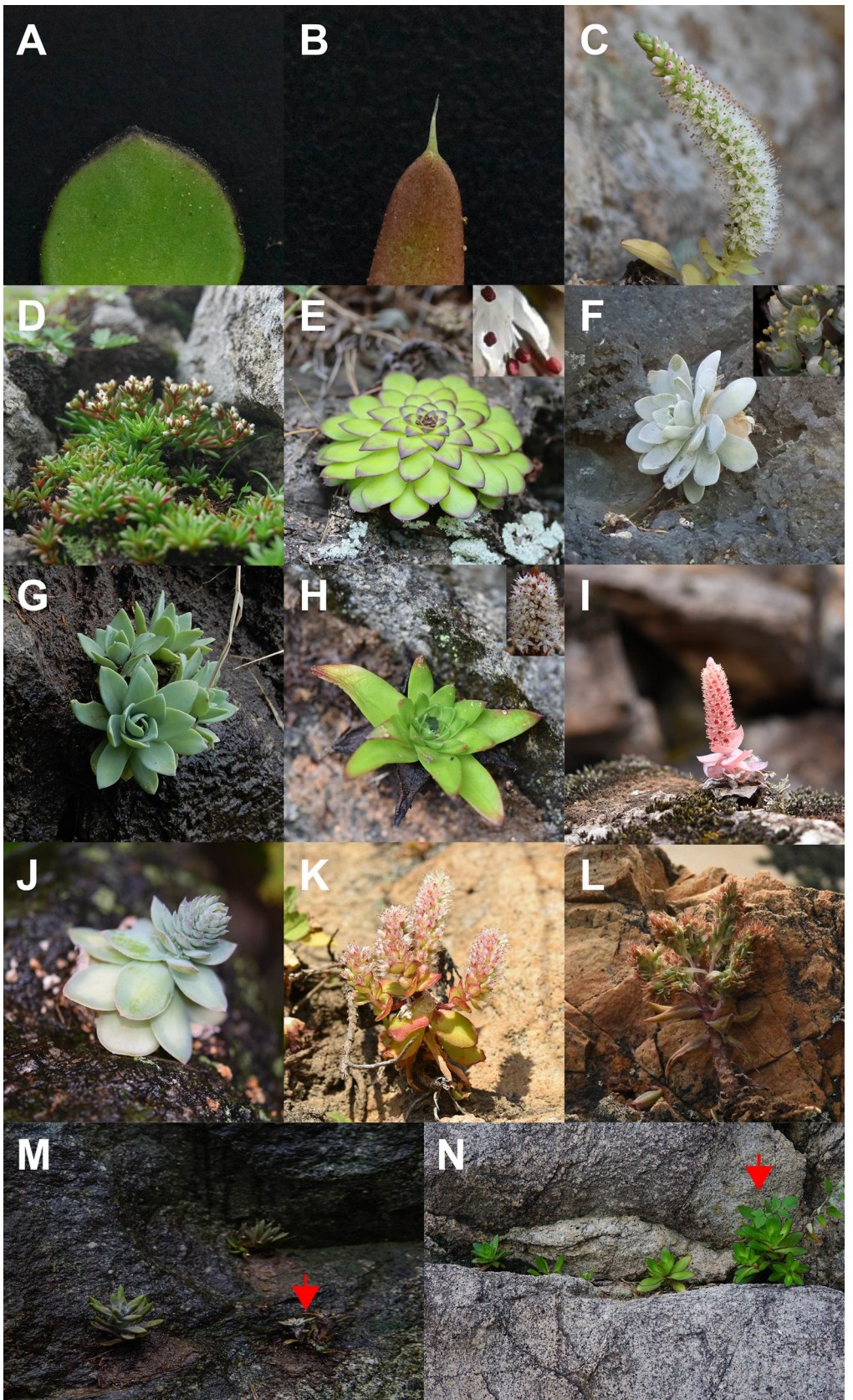

**Fig 6. Photographs of *Orostachys* species discussed in this study.** (A) absence and (B) presence of appendage such as a thorn at the leaf apex; (C) inflorescences of *Orostachys*; (D) *Meterostachys sikokianus*; (E) *O. margaritifolia*; (F) *O. iwarenge*; (G) *O. iwarenge* f. *magnus*; (H) O. *latielliptica*; O. *chongsunensis* in (I) sunny habitat and (J) shaded habitat; (K) *O. ramosa*; (L) *O. japonica* f. *polycephala*; individuals with branching at the base of the stem in (M) *O. japonica* and (N) *O, malacophylla* population.

*Orostachys* clade containing *O. iwarenge*, but in subsect. *Appeidiculatae*, characterized by the presence of an appendage at the leaf apex. This species was clearly distinguished from other taxa of this subsection due to morphological characteristics such as obovate leaves and purple anthers. Additionally, *O. iwarenge* f. *magnus* (Fig 6G), which has also been treated as a synonym for *O. iwarenge* (Fig 6F), was independently clustered, although the topology was different in each tree in this study. This species is distributed only on Ulleung-do Island, an oceanic island in Korea, and is reproductively completely isolated. Additionally, it is morphologically very similar to *O. iwarenge*, except for differences in leaf shape (oval in *O. iwarenge* f. *magnus* and oblong to spatulate in *O. iwarenge*) and stamen color (orange in *O. iwarenge* f. *magnus*, and yellow in *O. iwarenge*). Therefore, we strongly agree with Kim and Park [6] that these two species should be treated as independent taxa.

*O. latielliptica* (Fig 6H) was described as a new species [8] because it has appendages at the leaf apex, as well as glaucous ovate leaves, and one to four aggregated flowers on its pedicel. The latter two characteristics are unique and distinguish them from all other species. Therefore, the taxonomic position of this species as an independent taxon is supported based on these characteristics. However, the exact phylogenetic relationship could not be confirmed because the topology of this species was different in the two ML trees of this study (Fig 5). Furthermore, this species showed a very close relationship to *O. japonica*, which was treated as synonym [17], in the ML tree based on 79 protein coding gene sequences in this study. To investigate the exact taxonomic position of this species, further in-depth studies are necessary.

*O. chongsunensis* (Fig 6I and 6J), which was treated as a synonym for *O. japonica* by Ohba [17], was clustered within the subsect. *Orostachys* clade. This result was due to the absence of appendages at the leaf apex, indicating that *O. chongsunensis* is not the same species as *O. japonica*. In the ML trees in this study, *O. chongsunensis* showed a close relationship with *O. ramosa* (Fig 5A) and *O. malacophylla* (Fig 5B). Morphologically, this species was very similar to *O. malacophylla* except for purplish variegate leaves. Additionally, *O. malacophylla* was very similar to *O. ramosa* (Fig 6K) except that it does not branch at the base of the stem. In our experience, branching at the base of the stem is an occasional mutation in *O. malacophylla* (Fig 6N). Therefore, we concluded that the three species mentioned above are the same species; the purplish variegate leaves of *O. chongsunensis* are thought to be due to the growth environment and, specifically, factors such as exposure to limestone and the light intensity (Fig 6I and 6J). However, the exact phylogenetic relationship of *O. ramosa* could not be confirmed in this study because the topology in the two ML trees was different (Fig 5).

Meanwhile, since individuals with branching at the base of the stem in *O. japonica* are also relatively common in the natural population of *O. japonica* (Fig 6M), we concluded that *O. japonica* f. *polycephala* (Fig 6L) is an individual variation of *O. japonica*. Additionally, *O. japonica* f. *polycephala* showed the closest relationship to *O. japonica* in all ML trees in this study (Fig 5), which well supported our opinion.

## Conclusion

In this study, we assembled the chloroplast genomes of eight *Orostachys*, which had a total length ranging from 150,464 bp to 151,462 bp. The cp genomes of *Orostachys* and *Hylotelephium* had identical structures and were highly conserved. However, the structure of

*Meterostachys* was different due to the relatively large gene block (*trnS-GCU-trnS-GGA*) inversion, which is considered important information supporting its the taxonomic position as an independent genus. The results of phylogenetic analyses suggested that the two subsections of *Orostachys*, subsect. *Orostachys* and *Appendiculatae*, were independent genera. In addition, the results supported the taxonomic position of *O. margaritifolia* and *O. iwarenge* f. *magnus* as independent taxa. The results also suggested that *O. japonica* f. *polycephala* and *O. ramosa* were synonyms for *O. japonica* and *O. malacophylla*, respectively. Meanwhlie, the taxonomic position of *O. latielliptica* remains unclear. Also, it is still unknown whether *O. chongsunensis*, *O. malacophylla* and *O. ramosa*, are the same species.

## Supporting information

**S1 Fig. Comparison of thirteen chloroplast genome structure using MAUVE program.**
(PNG)

**S1 Table. Information for sample collection sites, voucher specimens, results of sequencing and genome assembly.**
(XLSX)

**S2 Table. The list and GenBank accession numbers used phylogenetic analyses in this study.**
(XLSX)

## Author Contributions

**Conceptualization:** Yoo-Jung Park, Kyeong-Sik Cheon.

**Data curation:** Ha-Rim Lee, Kyung-Ah Kim, Bo-Yun Kim.

**Formal analysis:** Bo-Yun Kim, Yoo-Jung Park, Yoo-Bin Lee.

**Funding acquisition:** Kyeong-Sik Cheon.

**Investigation:** Ha-Rim Lee, Yoo-Jung Park, Yoo-Bin Lee, Kyeong-Sik Cheon.

**Project administration:** Kyeong-Sik Cheon.

**Software:** Kyung-Ah Kim, Bo-Yun Kim, Yoo-Jung Park, Yoo-Bin Lee.

**Supervision:** Kyeong-Sik Cheon.

**Visualization:** Ha-Rim Lee, Kyung-Ah Kim, Bo-Yun Kim, Yoo-Jung Park, Yoo-Bin Lee.

**Writing – original draft:** Ha-Rim Lee, Kyung-Ah Kim, Kyeong-Sik Cheon.

**Writing – review & editing:** Kyung-Ah Kim, Kyeong-Sik Cheon.

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
