## [Decision Letter · Decision Letter 0]

23 Aug 2022

PONE-D-22-22176The complete chloroplast genome sequences of eight Orostachys species: comparative analysis and assessment of phylogenetic relationshipsPLOS ONE

Dear Dr. Cheon,

Thank you for submitting your manuscript to PLOS ONE. After careful consideration, we feel that it has merit but does not fully meet PLOS ONE’s publication criteria as it currently stands. Therefore, we invite you to submit a revised version of the manuscript that addresses the points raised during the review process.

We look forward to receiving your revised manuscript.

Kind regards,

Tzen-Yuh Chiang

Academic Editor

PLOS ONE

Journal Requirements:

2. Please ensure that you refer to Figure 4 in your text as, if accepted, production will need this reference to link the reader to the figure.

Reviewers' comments:

Reviewer's Responses to Questions

**Comments to the Author**

1. Is the manuscript technically sound, and do the data support the conclusions?

Reviewer #1: Yes

Reviewer #2: Yes

2. Has the statistical analysis been performed appropriately and rigorously? 

Reviewer #1: Yes

Reviewer #2: Yes

3. Have the authors made all data underlying the findings in their manuscript fully available?

Reviewer #1: Yes

Reviewer #2: Yes

4. Is the manuscript presented in an intelligible fashion and written in standard English?

Reviewer #1: Yes

Reviewer #2: Yes

5. Review Comments to the Author

Reviewer #1: This is a well-written manuscript. I suggest a minor revision for this ms. In the Nucleotide diversity among the Orostachys, pls add the result of rbcL and matK. Since rbcL and matK are the core DNA barcodes, both sequences could be retrieved from the chloroplast genome. It will be helpful to provide the relevant information. And I encourage the authors to submit their voucher photos and the sequences of rbcL and matK to the BOLD system, to facilitate the widespread use of their study results.

Reviewer #2: This study analyzed the complete chloroplast genomes of eight Orostachys species and compared these sequences with published sequences of the chloroplast genomes of the congeneric and closely related genera Meterostachys and Hylotelephium to assess the phylogenetic relationships between the two subfamilies of Orostachys and its closely related genera. To some extent, the taxonomic status of several taxa is clarified and the fact that taxonomists have different opinions on the classification of certain species is resolved. The techniques used in the study are reliable and the data support its conclusions, especially the phylogenetic analysis gives reliable results, and also well presented the shortcomings of the study and the direction of subsequent research, but the content part of the analysis of the chloroplast genome could be more complete.

In lines 314-315, “In our experience, branching at the base of the stem is an occasional mutation in O. malacophylla.” It is suggested here that it might be more rigorous to find literature or evidence for this rather than relying on experience, or it might be possible to describe clearly what experience is relied on，from which I think it is not very rigorous to conclude that O. malacophylla, O. ramosa, and O. chongsunensis are the same species.

Similarly, in lines 316-317, “the purplish variegate leaves of O. chongsunensis are thought to be due to the growth environment” providing the relevant basis here would make the conclusion more convincing.

It is strongly suggested to provide photographs of the studied species, especially those of the representative ones of each concerned group, for the readers’ easier understanding.

6. PLOS authors have the option to publish the peer review history of their article (what does this mean?). If published, this will include your full peer review and any attached files.

Reviewer #1: No

Reviewer #2: No

---

## [Author Response · Author response to Decision Letter 0]

4 Oct 2022

Response to Reviewers

We are pleased to resubmit for publication the revised version of PONE-D-22-22176 “The complete chloroplast genome sequences of eight Orostachys species: comparative analysis and assessment of phylogenetic relationships” We appreciated the constructive criticisms of the reviewers. We have addressed each of their concerns as outlined below.

→ We checked and reflected the template style of PLOS ONE at the request of the journal.

2. Please ensure that you refer to Figure 4 in your text as, if accepted, production will need this reference to link the reader to the figure.

→ Referenced Figure 4 in the manuscript.

→ References have been reviewed and revised once again.

Reviewer 1.

This is a well-written manuscript. I suggest a minor revision for this ms. In the Nucleotide diversity among the Orostachys, pls add the result of rbcL and matK. Since rbcL and matK are the core DNA barcodes, both sequences could be retrieved from the chloroplast genome. It will be helpful to provide the relevant information. And I encourage the authors to submit their voucher photos and the sequences of rbcL and matK to the BOLD system, to facilitate the widespread use of their study results.

→ We added the result of nucleotide diversity of rbcL and matK. We are trying to upload the sequences of rbcL and matK to the BOLD system, but it is not completed at this time. We promise to upload as soon as possible.

Reviewer 2.

This study analyzed the complete chloroplast genomes of eight Orostachys species and compared these sequences with published sequences of the chloroplast genomes of the congeneric and closely related genera Meterostachys and Hylotelephium to assess the phylogenetic relationships between the two subfamilies of Orostachys and its closely related genera. To some extent, the taxonomic status of several taxa is clarified and the fact that taxonomists have different opinions on the classification of certain species is resolved. The techniques used in the study are reliable and the data support its conclusions, especially the phylogenetic analysis gives reliable results, and also well presented the shortcomings of the study and the direction of subsequent research, but the content part of the analysis of the chloroplast genome could be more complete.

In lines 314-315, “In our experience, branching at the base of the stem is an occasional mutation in O. malacophylla.” It is suggested here that it might be more rigorous to find literature or evidence for this rather than relying on experience, or it might be possible to describe clearly what experience is relied on，from which I think it is not very rigorous to conclude that O. malacophylla, O. ramosa, and O. chongsunensis are the same species.

→ We have added photos (Fig 6) to support our experience.

Similarly, in lines 316-317, “the purplish variegate leaves of O. chongsunensis are thought to be due to the growth environment” providing the relevant basis here would make the conclusion more convincing.

→ We have also added photos (Fig 6) to support this.

It is strongly suggested to provide photographs of the studied species, especially those of the representative ones of each concerned group, for the readers’ easier understanding.

→ We have added various photos (Fig 6) related to this study.

---

## [Decision Letter · Decision Letter 1]

28 Oct 2022

The complete chloroplast genome sequences of eight Orostachys species: comparative analysis and assessment of phylogenetic relationships

PONE-D-22-22176R1

Dear Dr. Cheon,

We’re pleased to inform you that your manuscript has been judged scientifically suitable for publication and will be formally accepted for publication once it meets all outstanding technical requirements.

Kind regards,

Tzen-Yuh Chiang

Academic Editor

PLOS ONE

Additional Editor Comments (optional):

Reviewers' comments:

Reviewer's Responses to Questions

**Comments to the Author**

1. If the authors have adequately addressed your comments raised in a previous round of review and you feel that this manuscript is now acceptable for publication, you may indicate that here to bypass the “Comments to the Author” section, enter your conflict of interest statement in the “Confidential to Editor” section, and submit your "Accept" recommendation.

Reviewer #1: All comments have been addressed

Reviewer #2: All comments have been addressed

2. Is the manuscript technically sound, and do the data support the conclusions?

Reviewer #1: Yes

Reviewer #2: Yes

3. Has the statistical analysis been performed appropriately and rigorously? 

Reviewer #1: Yes

Reviewer #2: N/A

4. Have the authors made all data underlying the findings in their manuscript fully available?

Reviewer #1: Yes

Reviewer #2: Yes

5. Is the manuscript presented in an intelligible fashion and written in standard English?

Reviewer #1: Yes

Reviewer #2: Yes

6. Review Comments to the Author

Reviewer #1: My questions were well answered. I am satisfied with the revision. Hope the voucher photoes were upload to BOLD system later.

Reviewer #2: (No Response)

7. PLOS authors have the option to publish the peer review history of their article (what does this mean?). If published, this will include your full peer review and any attached files.

Reviewer #1: No

Reviewer #2: No

---

## [Editor Report · Acceptance letter]

2 Nov 2022

PONE-D-22-22176R1 

The complete chloroplast genome sequences of eight *Orostachys* species: comparative analysis and assessment of phylogenetic relationships 

Dear Dr. Cheon:

I'm pleased to inform you that your manuscript has been deemed suitable for publication in PLOS ONE. Congratulations! Your manuscript is now with our production department. 

Kind regards, 

on behalf of

Dr. Tzen-Yuh Chiang 

Academic Editor

PLOS ONE